# Network Analysis of a Membrane-Enriched Brain Proteome across Stages of Alzheimer’s Disease

**DOI:** 10.3390/proteomes7030030

**Published:** 2019-08-27

**Authors:** Lenora Higginbotham, Eric B. Dammer, Duc M. Duong, Erica Modeste, Thomas J. Montine, James J. Lah, Allan I. Levey, Nicholas T. Seyfried

**Affiliations:** 1Department of Neurology, Emory University School of Medicine, Atlanta, GA 30322, USA; 2Department of Biochemistry, Emory University School of Medicine, Atlanta, GA 30322, USA; 3Department of Pathology, Stanford University School of Medicine, Stanford, CA 94305, USA

**Keywords:** synapse, vesicles, proteomics, preclinical, biomarkers

## Abstract

Previous systems-based proteomic approaches have characterized alterations in protein co-expression networks of unfractionated asymptomatic (AsymAD) and symptomatic Alzheimer’s disease (AD) brains. However, it remains unclear how sample fractionation and sub-proteomic analysis influences the organization of these protein networks and their relationship to clinicopathological traits of disease. In this proof-of-concept study, we performed a systems-based sub-proteomic analysis of membrane-enriched post-mortem brain samples from pathology-free control, AsymAD, and AD brains (*n* = 6 per group). Label-free mass spectrometry based on peptide ion intensity was used to quantify the 18 membrane-enriched fractions. Differential expression and weighted protein co-expression network analysis (WPCNA) were then used to identify and characterize modules of co-expressed proteins most significantly altered between the groups. We identified a total of 27 modules of co-expressed membrane-associated proteins. In contrast to the unfractionated proteome, these networks did not map strongly to cell-type specific markers. Instead, these modules were principally organized by their associations with a wide variety of membrane-bound compartments and organelles. Of these, the mitochondrion was associated with the greatest number of modules, followed by modules linked to the cell surface compartment. In addition, we resolved networks with strong associations to the endoplasmic reticulum, Golgi apparatus, and other membrane-bound organelles. A total of 14 of the 27 modules demonstrated significant correlations with clinical and pathological AD phenotypes. These results revealed that the proteins within individual compartments feature a heterogeneous array of AD-associated expression patterns, particularly during the preclinical stages of disease. In conclusion, this systems-based analysis of the membrane-associated AsymAD brain proteome yielded a unique network organization highly linked to cellular compartmentalization. Further study of this membrane-associated proteome may reveal novel insight into the complex pathways governing the earliest stages of disease.

## 1. Introduction

Alzheimer’s disease (AD) is characterized by an early, asymptomatic phase (AsymAD) in which individuals exhibit AD neuropathology in the absence of clinically detectable cognitive decline [1,2,3,4,5,6]. This preclinical stage of disease presents a critical window for early detection and intervention. Yet, much regarding this early phase and its underlying biological mechanisms remain unclear. Systems-level analysis has emerged as a useful tool for the large-scale investigation of such disease-related biology. Initially applied to the analysis of transcriptomes, algorithms such as weighted gene co-expression network analysis (WGCNA) allowed for the classification of these complex datasets into meaningful modules of co-expressed genes linked to specific cell types, organelles, and biological pathways [7,8].

We have previously implemented this systems-based approach in proteomic analysis and demonstrated its utility in identifying altered networks of protein co-expression in the AsymAD brain [9]. Using post-mortem cortical samples from control, AsymAD, and AD subjects, we identified disease-specific modules of co-expressed proteins, several of which demonstrated notable changes in preclinical disease. These disease-associated co-expression modules were preserved across different AD cohorts and mapped strongly to cognitive status and neuropathology [9]. In addition, many were enriched with markers linked to specific cell types, including neurons, oligodendrocytes, astrocytes, and endothelial cells [9,10]. We have since applied this algorithm to other brain-derived proteomic datasets to explore additional questions surrounding AD progression [10,11,12,13]. Yet, while this multi-network analytical approach is proving a promising proteomic tool, we have applied it only to unfractionated brain samples and have yet to explore its utility in a sub-fractionated proteome. Indeed, it remains unclear as to whether applying this algorithm to a less complex proteomic sample would yield modules with similarly strong cell type and disease phenotype associations. Furthermore, it is unknown whether such an analysis could offer new or otherwise valuable insights into the systems-based underpinnings of preclinical AD and disease progression.

In this proof-of-concept study, we investigated these questions by applying a network-based approach to membrane-enriched post-mortem brain samples of AsymAD and symptomatic disease. We specifically chose a membrane fractionation protocol due to our interest in exploring the behavior of synaptic and other cell signaling proteins in preclinical disease. Our prior work has suggested that despite its well-established correlations with cognitive decline, synaptic dysfunction may begin during the pre-symptomatic stages of AD [9]. In addition, we have found that variation in synaptic protein abundance may contribute to cognitive resilience during the aging process [11]. These findings have underscored the potential role that synaptic proteins may have as early AD diagnostic or therapeutic targets. Ultimately, our study generated a proteome derived from not only the synapse, but multiple other membrane-bound cellular compartments. Systems-based analysis revealed highly organelle-specific modules, unique to those of the unfractionated AD brain, with strong correlations to clinicopathological AD phenotypes. Overall, these results indicate that further network-based study of the membrane-enriched AD proteome may provide novel insight into the protein associations governing disease pathogenesis and progression.

## 2. Materials and Methods

### 2.1. Case Selection

All brain tissue used in this analysis was derived from the autopsy collection of the Adult Changes In Thought (ACT) cohort, in which participants were randomly sampled from a large health management organization in King County, Washington, and subjected to serial cognitive screening every two years using the Cognitive Assessment Screening Instrument (CASI) [14,15]. Prior to enrolling in the ACT study, all individuals provided written informed consent and the University of Washington and Group Health Cooperative of Puget Sound institutional review boards (IRB) reviewed and approved the study. Post-mortem neuropathological evaluation of amyloid plaque distribution was performed according to the Consortium to Establish a Registry for Alzheimer’s Disease (CERAD) criteria [16], while extent of spread of neurofibrillary tangle pathology was assessed in accordance with the Braak staging system [17]. Eighteen cases were selected for proteomic analysis and sorted into the following three groups: (i) cognitively intact individuals without AD pathology (controls), (ii) cognitively intact individuals with AD pathology (AsymAD), and (iii) symptomatic individuals with AD pathology. The latter symptomatic group all exhibited mild-to-moderate cognitive deficits on the CASI. The inclusion criteria for each cohort are outlined in Appendix A. All 18 selected cases were matched according to age at death. To limit potential confounders, we also specifically selected cases with minimal coexisting neuropathology, such as Lewy bodies and prior infarcts.

### 2.2. Membrane Enrichment

All tissue samples were derived from the middle frontal gyrus, corresponding to Brodmann areas 8 and 9, with minimal inclusion of white matter. This region was selected because it demonstrates cortical thinning during preclinical AD and its CERAD scores tend to mirror the brain as a whole [18]. The membrane-enrichment strategy employed was modified from previously published methods [19,20,21]. In brief, frozen tissue (200 ± 20 mg) was first homogenized in a low salt, buffered sucrose solution (0.24 M sucrose, 25 mM NaCl, 50 mM HEPES pH 7.0, 1 mM EDTA) with protease and phosphatase inhibitors. Total tissue homogenate (H) was centrifuged for 10 min at 1500× *g* (Eppendorf 5417C) and the supernatant (S1′) removed. The remaining pellet (P1′) was resuspended in sucrose buffer and again centrifuged for 10 min at 1500× *g.* This supernatant (S1”) was combined with S1′ to generate S1. The remaining pellet (P1), comprised largely of unhomogenized tissue, large cellular debris, and nuclear components, was stored at −80 °C. S1 was then centrifuged at 180,000× *g* for one hour at 4 °C (Beckman Optima TLX ultracentrifuge, Ramsey, MN, TLA 100.4 rotor). Afterward, the supernatant (S2), comprised of cytosol-enriched sample, was removed and stored at −80 °C. The resulting pellet (P2) was resuspended in 1 mL of 0.1 sodium carbonate pH 11 with protease and phosphatase inhibitors (Sigma Aldrich) and sonicated (Sonic Dismembrator, Fisher Scientific, Waltham, MA, USA) three times for five seconds each at 20% amplitude (maximum intensity). The sonicated sample was then centrifuged at 180,000× *g* for one hour at 4 °C. The supernatant (S3) was removed and stored at −80 °C and the resulting membrane-enriched pellet (P3) was dissolved in 100 µL 8M urea to generate the final membrane fraction (M). Urea is a well-known chaotropic agent capable of weakening hydrophobic interactions between insoluble proteins [22]. Therefore, its use to dissolve the final M fractions served to optimally free hydrophobic membrane-associated proteins and increase the ease of subsequent gel digestion and proteomic measurement. The H, S1, S2, and M fractions of an individual control case were analyzed by silver stain, as previously described [19] (Appendix A). Briefly, protein (1 μg) from each fraction was separated on a 10% SDS gel. To ensure equal loading, protein concentrations of all fractions were determined by the bicinchoninic acid (BCA) method (Pierce, Rockford, IL, USA). The gel was then fixed in a solution containing 50% methanol and 5% acetic acid for 10 min. After a brief wash in deionized water, the gel was rinsed in 0.02% sodium thiosulfate for 1 min, stained with 0.1% silver nitrate for 10 min, and developed in a solution of 3% sodium carbonate and 0.05% formaldehyde until protein bands were sufficiently stained. All 18 membrane-enriched fractions were also analyzed using this silver staining protocol (Appendix A).

### 2.3. Mass Spectrometry Based Proteomics

In preparation for LC-MS/MS analysis, individual membrane (M) fractions (20 µg) from each case including internal standards was reduced with 5 mM dithiothreitol (DTT) for 15 min at 37 °C and then alkylated with 20 mM iodoacetamide (IAA) for 30 min at 37 °C in the dark [23]. The alkylated samples were separated on a 10% SDS gel and stained with Coomassie Blue G-250. Each sample lane was cut into five gel bands corresponding to molecular weight ranges, in order to increase the depth of coverage of the proteome (Appendix A). The gel pieces were then digested overnight in 12.5 µg/mL trypsin at 37 °C. Subsequently, the samples were extracted in a solution of 5% formic acid and 50% acetonitrile (ACN).

The resulting peptides were analyzed by high resolution LC-MS/MS as essentially described by the authors of [23]. An equal amount of each peptide sample was resuspended in loading buffer, which was comprised of 0.1% formic acid, 0.03% trifluoroacetic acid, and 1% acetonitrile. The samples were then loaded onto a 20 cm nano-LC column (internal diameter 100 µm) packed with Reprosil-Pur 120 C18-AQ 1.9 µm beads (Dr. Maisch GmbH) and eluted over 1 h with 4–80% buffer B reverse phase gradient (Buffer A: 0.1% formic acid, 1% acetonitrile in water; Buffer B: 0.1% formic acid in acetonitrile) generated by a NanoAcquity UPLC system (Waters Corporation). Peptides were ionized with 2.0 kV electrospray ionization voltage from a nano-ESI source (Thermo) on a hybrid LTQ XL Orbitrap mass spectrometer (Thermo Finnigan, San Jose, CA, USA). Data dependent acquisition of centroid MS spectra at 30,000 resolution and MS/MS spectra were obtained in the LTQ following collision induced dissociation (collision energy 35%, activation Q 0.25, activation time 30 ms) for the top 10 precursor ions with charge determined by the acquisition software to be *z* ≥ 2. The SageN Sorcerer SEQUEST 3.5 algorithm was used to search and match MS/MS spectra to a complete semi-tryptic human proteome database (NCBI reference sequence revision 50, with 66,652 entries) including pseudo-reversed decoy sequences [24,25] and the common repository of adventitious proteins (cRAP version 2012.01.01) with a 20 ppm mass accuracy threshold. Only *b* and *y* ions were considered during the database match. In addition, Xcorr and ΔCn were dynamically increased for groups of peptides organized by a combination of trypticity (fully or partial) and precursor ion charge state to remove false positive hits and decoys until achieving a false discovery rate (FDR) of < 1%. Searching parameters included precursor ion mass tolerance (20 ppm), partial tryptic restriction, fixed mass shift for modification of carboxamidomethylated Cys (+57.0215 Da) and dynamic mass shift for oxidized Met (+15.9949). Peptide quantification was performed based on the extracted ion current (XIC) measurements of identified peptides [21,26]. Ion intensities for identified peptides were extracted in full-MS survey scans of high-resolution and a ratio of the peak intensities for the peptide precursor ion was calculated using in-house software as previously published [21,26,27,28]. Accurate peptide mass and retention time (RT) was used to derive signal intensity for every peptide across LC-MS/MS runs for each case. For those proteins identified by ≥3 peptides, we averaged the extracted ion intensities for the three most intense tryptic peptides, which yields an abundance measurement for each identified protein with a coefficient of variation (CV) less than ±10% across technical replicates [29].

### 2.4. Differential Expression

Bootstrap non-parametric regression of the protein intensity matrix was performed using a model incorporating case status and case covariates for age, gender, and postmortem interval (PMI) [9,13]. We regressed for PMI, as it has previously been shown that this interval may influence protein levels [30]. Yet, it is notable that the post-mortem intervals of the cases included in this study were of very short duration, ranging from 2.5 to 8.5 h. In addition, the average intervals for each of the three cohorts were similar (3.8 to 4.7 h). Following regression, differentially expressed proteins were then identified using one-way ANOVA followed by Tukey’s post-hoc test for pairwise comparisons in R statistical software as previously described [9]. Three pairwise comparisons were considered in this analysis, including (i) controls vs. AsymAD, (ii) controls vs. AD, and (iii) AsymAD vs. AD. Proteins with a Tukey pairwise comparison *p* value below 0.05 were considered significantly altered.

### 2.5. Weighted Protein Correlation Network Analysis (WPCNA)

A weighted protein co-expression network was built using the above post-regressed protein abundance values using blockwiseModules WGCNA function (WGCNA 1.47 R package) with the following parameters: soft threshold power beta = 11.5, deepSplit = 4, minimum module size of 20, TOMdenom = ”mean”, corType = ”bicor”, merge cut height of 0.07, signed network with partitioning about medioids respecting the dendrogram, and a reassignment threshold of *p* = 0.05. The resulting 27 modules or groups of co-expressed proteins were used to calculate module eigenproteins as previously described [9]. Pearson correlations between each protein and each module eigenprotein were performed; this module membership measure is defined as kME and is provided in Appendix A. Module eigenproteins were correlated with a variety of AD-associated phenotypes (i.e., AD diagnosis, cognitive scores, and levels of amyloid and tau burden) using biweight midcorrelation (bicor) analysis.

### 2.6. Cell Type Enrichment

Cell type enrichment of the WPCNA modules was assessed as previously described [9]. Briefly, the corresponding gene symbols of each module were cross-referenced with lists of genes known to be preferentially expressed in different cell types. Significance of cell type enrichment within each module was then determined using a one-tailed Fisher’s exact test and corrected for multiple comparisons by the FDR (Benjamini–Hochberg) method.

### 2.7. Gene Ontology (GO) Enrichment

Functional enrichment within the WPCNA modules was determined using the GO-Elite v1.2.5 python package [31] and Ensembl v62 mart database for *Homo sapiens.* The corresponding gene symbols of each protein module were analyzed for over-representation of human gene ontologies within this database related to biological processes, molecular functions, and cellular compartments. Significance of ontology enrichment within each module was determined using a one-tailed Fisher’s exact test and corrected for multiple comparisons by the FDR (Benjamini–Hochberg) method. Ontologies of interest (*p* < 0.05) within each module were manually curated and reported as described in the results.

### 2.8. Immunoblotting

Equal amounts of each sample were loaded onto a 10% SDS gel. To ensure equal loading, protein concentration was determined by BCA method. Separated proteins were transferred onto PVDF Immobilon-P membranes (Millipore, Billerica, MA, USA) overnight at 4 °C. Blots were subsequently blocked for 2 h at room temperature, probed with primary antibody overnight at 4 °C, and incubated in the dark for 1 h at room temperature with fluorophore-conjugated secondary antibodies (1:20,000). All blots were scanned and quantified with an Odyssey Infrared Imaging System (Li-Cor Biosciences, Lincoln, NE, USA). Primary antibodies used in this study included Synaptophysin (1:1000, mouse monoclonal; Boehringer); GAP43 (1:1000, rabbit polyclonal; Abcam, Cambridge, MA, USA); phospho S41 GAP43 (1:1000 rabbit monoclonal; Abcam, Cambridge, MA, USA); and β-Actin (1:1000 goat polyclonal; Abcam, Cambridge, MA, USA). All antibody dilutions noted above reflect prior dilution of each antibody (1:1) with glycerol.

### 2.9. Over-Representation Analysis for Unfractionated and Membrane Protein Networks

The unfractionated network used in this analysis was recently published and described in detail [9]. This published network, comprised of control, AsymAD, and AD cases derived from the Baltimore Longitudinal Study of Aging (BLSA), was chosen for over-representation analysis because it was generated using similar label-free LC-MS/MS quantitation and WPCNA methods. Furthermore, we have previously demonstrated preservation of its modules across other unfractionated AD and neurodegenerative cases. The over-representation analysis was performed using a one-sided Fisher exact test with 95% confidence intervals calculated according to the R function fisher.test, as previously described, but with an alternative hypothesis parameter set to “greater” [9,13]. FDR adjusted *p*-values from these hyper-geometric test comparisons were used in order to reduce false positives.

### 2.10. Data and Software Availability

All raw proteomic data generated contributing to the described work will be deposited electronically on the PRoteomics IDEntification (PRIDE) Archive Database (https://www.ebi.ac.uk/pride/archive) at project accession PXD014376. Specific software will also be made available upon request.

## 3. Results

### 3.1. Brain Fractionation Demonstrates Membrane Protein Enrichment

We and others have shown that membrane and synaptic-rich fractions can be successfully derived from post-mortem brain tissues [19,32,33,34]. As previously described, our membrane-enrichment strategy generates a fraction with as much as 2.5-fold enrichment of proteins associated with the cell surface and organelle membranes, e.g., mitochondria, transport vesicles, endoplasmic reticulum, and synapses [19,20,21]. In the current study, we applied this membrane-enrichment protocol to eighteen individual cases representing the following three groups: (i) pathology-free, cognitively normal individuals (i.e., controls); (ii) cognitively normal individuals with pathological amyloid plaque burden (i.e., AsymAD); and (iii) cognitively impaired individuals with pathological amyloid plaque burden (i.e., AD). The dorsolateral prefrontal cortex (DLPFC, Brodmann area 9) was analyzed because it demonstrates cortical thinning during preclinical AD, and amyloid deposition in this region tends to mirror deposition in the brain as a whole [18]. The three cohorts were matched for age at death. Aside from associated neurofibrillary tangles, the AsymAD and AD cases had minimal levels of comorbid neuropathology (Appendix A). Despite harboring moderate amounts of plaque and tangle pathology, the AsymAD cases featured cognitive scores comparable to controls at time of death.

Following membrane-fractionation, we employed various strategies to examine the success of our protocol in the current samples. Using gene ontology (GO) protein classification and the ratios of peptide spectral counts in our membrane and soluble fractions, we demonstrated that over 90% of transmembrane and over 60% of membrane-associated proteins were enriched in our final membrane fractions (Figure 1A). In addition, immunoblotting the sample fractions of 6 independent controls for the membrane-associated protein neuromodulin (GAP43) revealed its significant enrichment in the final membrane fractions, as compared to the total homogenate and soluble fractions (Figure 1B). An even higher enrichment of phosphorylated (pSer41) GAP43 was observed, which is consistent with the established role that phosphorylation at Ser41 plays in targeting GAP43 to membranes [35]. In summary, this fractionation approach successfully enriched our samples with membrane-associated proteins, including known synaptic markers.

### 3.2. Proteomic Analysis Reveals Differential Protein Abundance Across Alzheimer’s Disease Stages

The membrane fraction from each case was resolved by SDS-PAGE and in-gel digested from five molecular weight regions. The resulting peptides were analyzed by liquid chromatography mass spectrometry (LC-MS/MS) and identified proteins were subsequently quantified for each case based on peptide ion intensities [26]. Using this label-free quantification approach, we identified and quantified a total of 16,310 peptides from 1808 protein groups measured by at least one unique peptide and two peptide spectral matches (Appendix A). These proteins mapped to 1785 unique gene symbols across the 18 case samples. Appendix A also provides the relative abundances for all membrane fraction proteins identified with peptide counts and percent coverage.

As expected, relative label-free quantification of amyloid precursor protein (APP) correlated strongly with CERAD scores (r = 0.77) (Figure 2A). The AsymAD cases generally demonstrated greater APP levels than the controls, while the AD cohort yielded the highest APP abundances. APP can be a direct precursor to amyloid beta (Aβ). For example, the APP tryptic peptide we quantified in our samples mapped directly to Aβ residues 17–28 and thus could be used as a surrogate for amyloid levels in the sample [9]. However, the quantified APP does not differentiate between full-length APP and the cleaved Aβ species found in amyloid plaques. Furthermore, given our method of tissue fractionation, the identified APP peptide likely represents membrane-associated intraneuronal/vesicular Aβ as opposed to the insoluble Aβ of neuritic plaques typically best isolated by detergent extraction techniques [27]. These factors may account for the variability of APP levels identified among the AsymAD cohort despite their similar CERAD scores.

A total of 530 unique proteins demonstrated significantly altered expression levels across the following three comparisons: (i) controls vs. AsymAD (*n* = 106), (ii) controls vs. AD (*n* = 279), and (iii) AsymAD vs. AD (*n* = 348) (Figure 2B,C). Overall, our analysis revealed a notable degree of altered protein expression across all stages of the disease continuum. APP and the synaptic protein SNAP25 were the only two proteins to demonstrate statistically significant expression changes between all three groups. Though, in contrast to APP, SNAP25 levels decreased throughout the course of disease, in accordance with prior literature [36,37] (Figure 2B,C). Appendix A provides the ANOVA and Tukey post-hoc pairwise comparison *p* values for the 1808 proteins included in this analysis.

### 3.3. Protein Co-Expression Network Analysis Yields Modules Organized by Membrane-Associated Cellular Compartments

Weighted protein co-expression network analysis (WPCNA) defines biologically meaningful modules of proteins based on co-expression patterns in large-scale proteomic studies [7,38,39,40,41]. Defining protein co-expression patterns is particularly effective at linking groups of similarly expressed proteins with clinical and pathological phenotypes [42]. We applied WPCNA to all 1808 proteins quantified across the analyzed cases. Protein abundance values were adjusted for influences of age, sex, and post-mortem interval (PMI) [9,10,11,12]. A total of 27 modules (M) of co-expressed proteins were identified and ranked by size, ranging from M1 (largest, 264 proteins) to M27 (smallest, 21 proteins). As shown in Figure 3A, WPCNA results in a dendrogram in which modules with similar expression patterns cluster near each other. The expression profile of each module is represented by its calculated eigenprotein, as previously described [9]. Briefly, an eigenprotein is defined as the first principal component of a given module that serves as a representative, weighted expression profile for that module. Appendix A provides a module membership for each protein in the network.

In our analysis of the unfractionated proteome, we found that cell type specificity played a significant role in module composition [9], suggesting that cellular changes in abundance and phenotype could be one of the biggest drivers of protein co-expression in the AD brain. However, in this membrane-fractionated proteome, cell type specificity played a very minimal role in the module composition. We evaluated cell type association for each module by cross-referencing its member proteins against lists of proteins known to be enriched in isolated neurons and glial cells [43]. A one-tailed Fisher’s exact test was then applied to determine statistically significant levels of marker protein enrichment within each module. We ultimately discovered that only 2 of our 27 modules were enriched with cell-specific markers (Figure 3A). Module 6 (M6) contained enrichment of proteins found in oligodendrocytes, including the myelin-associated molecules CNP, MAG, and PLP1. On the other hand, M21 was enriched with neuronal markers. Its hub proteins included multiple members of the Ca^2+^/calmodulin-dependent protein kinase subfamily (i.e., CAMK2G, CAMK2A, CAMK2B), which regulate synaptic calcium signaling [44].

To further investigate the biological factors influencing our module composition, we subsequently applied a gene ontology (GO) analysis. In this analysis, the corresponding gene symbols of each protein module were analyzed for over-representation of human gene ontologies related to biological processes, molecular functions, and cellular compartments. These results revealed that nearly all 27 modules demonstrated very strong relationships with distinct cellular compartments (Figure 3B). In fact, several modules, such as M15, M16, and M20, were defined almost solely by their cellular compartmentalization as opposed to functional associations. Most modules localized strongly to membrane-bound compartments, while a small number (*n* = 5) were associated with the cytosol or cytoskeleton. A wide variety of membrane-bound compartments were represented among our networks. Of these, the mitochondrion was significantly associated with the greatest number of modules (*n* = 6), followed by the cell surface compartment (*n* = 4) and endoplasmic reticulum (ER) (*n* = 3). Interestingly, there were certain modules that demonstrated significant associations with more than one compartment, potentially underscoring the familiar concept of interorganellar communication [45]. For instance, M2 was significantly associated with the membranes of both the mitochondrion and ER. In addition, M8 localized strongly not only to the cell surface, but also to the Golgi apparatus. It should be noted that within the cell surface networks, modules heavily associated with the synapse, such as M5 and M8, were also strongly linked to less specific terms such as “plasma membrane” and “integral to plasma membrane”. This indicated that these modules contained surface proteins across a variety of cortical cell types, which likely accounted for their lack of significant enrichment with neuronal markers.

These compartment-driven groupings in many instances aligned with the clusters defined in the initial WPCNA eigenprotein dendrogram. However, the ontology analysis did at times group together modules that were quite removed from each other in the WPCNA network. This suggested that within individual cellular compartments, there existed groups of proteins with markedly different expression patterns in diseased subjects. For instance, in the WPCNA dendrogram, M26 was far removed from the other mitochondrial modules, representing a set of proteins with a highly unique expression pattern compared to other modules in its compartment. Accordingly, we later found that M26 demonstrated stable levels in control and diseased individuals, diverging from the other mitochondrial modules, which all decreased in abundance among diseased cases. Yet, even among these decreasing mitochondrial modules, we discovered more subtle differences in preclinical expression patterns, as detailed later in the Results section. In summary, these results indicated that our approach is effectively able to identify cell type-independent variation in the protein expression patterns within cellular compartments.

### 3.4. Overlap Analysis Demonstrates Differences in Module Composition between Unfractionated and Membrane-Associated AD Networks

The compartment-driven modules of our membrane-fractionated proteome appeared to diverge significantly from the cell type-specific modules derived in our prior unfractionated brain analyses [9,10]. To further assess these differences in module composition, we used an over-representation analysis (ORA) to relate modules across fractionated and unfractionated AD brain networks. In this ORA, we included a recently published network analysis of 97 unfractionated cortical samples from healthy control, AsymAD, and AD cases derived from the Baltimore Longitudinal Study of Aging (BLSA) [9]. As described previously [9], label-free LC-MS/MS quantitation and subsequent WPCNA of these cortical samples yielded a co-expression network comprised of 2735 proteins and 16 co-expression modules. These modules were highly preserved across different cohorts of unfractionated brain tissues derived from both AD and other neurodegenerative cases. Indeed, an ORA of this BLSA network and that of a separate Emory cohort of unfractionated degenerative cases demonstrated significant overlap of nearly all modules (14/16; 87.5%) between the datasets [9].

In contrast, the ORA between this unfractionated BLSA network and that of our membrane-fractionated proteome revealed that only 37% (10/27) of membrane modules significantly overlapped between datasets (Figure 4). Likewise, only 6 of the 16 BLSA modules (38%) overlapped in the membrane-fractionated network. Of these 6 cognate BLSA modules generated from unfractionated (U) brain tissues, the three largest (U-M1, U-M2, and U-M3) accounted for the majority of overlap. U-M1, which corresponded strongly to neuronal-specific markers and synaptic transmission ontology, demonstrated statistically significant overlap (*p* < 0.05; FDR < 0.05) with membrane-associated (M) modules M-M8, M-M9, and M-M21. Accordingly, both M-M8 and M-M9 were comprised of proteins strongly associated with the cell surface compartment. While M-M21 correlated most significantly to the ER, it also demonstrated weakly positive associations with cell surface/synaptic ontology. In addition, M-M21 was the only module of the membrane-fractionated network significantly enriched with neuronal markers. The second-largest unfractionated module, U-M2, strongly correlated to oligodendrocyte markers and myelination. U-M2 overlapped most significantly with the proteasome-associated M-M6, suggesting that myelination may account for much of the protein turnover in the AD cortex. Meanwhile, the mitochondrion-linked U-M3 highly overlapped with membrane-associated modules M-M4, M-M15, and M-M19, all similarly correlated to the mitochondrial compartment (Figure 3). Interestingly, M-M2 and M-M26 did not overlap with U-M3 despite their similarly strong mitochondrial associations, suggesting that the membrane-associated proteome may offer a more complex window into mitochondrial protein co-expression in the AD brain. A final notable overlap occurred between the small unfractionated module U-M16 and the membrane-associated M-M18, both of which significantly corresponded to ribosomes and the ribonucleoprotein complex.

The vast amount of non-overlap between the networks signified substantial differences in the module composition between the two datasets. The non-overlapping portion of the unfractionated proteome included several large glia-enriched modules linked strongly to cytoplasmic structures or processes, including U-M5 (astrocyte/microglia-enriched; extracellular matrix), U-M6 (astrocyte/microglia-enriched; inflammatory response), and U-M9 (astrocyte-enriched; oxidoreductase activity). The remaining non-overlapping modules of the unfractionated dataset included U-M10, a module strongly correlated to DNA/RNA binding and heavily comprised of nucleoplasm proteins depleted in our fractionation protocol. In addition, protein-folding regulation, heavily represented in the unfractionated proteome by U-M7 (“de novo” protein folding) and U-M11 (unfolded protein binding), was conspicuously absent among membrane-associated modules. Meanwhile, protein import and targeting appeared to be a more prominent component of the membrane proteome, as represented by the non-overlapping M-M3 and its top functional ontologies. Another notable non-overlapping membrane module was the cell surface-associated M-M5. Despite its links to synaptic ontologies, M-M5 diverged from its fellow surface modules M-M8 and M-M9 in its failure to overlap with the synapse-associated U-M1. Accordingly, as outlined below, we ultimately found that M-M5 demonstrated a markedly different expression pattern throughout AD compared to the other surface-associated membrane modules. Overall, these results indicated that our compartment-driven membrane network was indeed unique in many aspects of its module composition when compared to the unfractionated AD network.

### 3.5. Membrane-Derived Modules Demonstrate Links to Clinical and Pathological Phenotypes of Alzheimer’s Disease

To determine if the membrane-fractionated networks we resolved had associations with clinicopathological disease phenotypes, we performed a biweight midcorrelation (bicor) analysis of each module with AD diagnosis, cognitive scores, and levels of amyloid and tau burden. Of the 27 modules identified in this proteome, 14 demonstrated significant correlations to clinical or pathological phenotypes of disease (Figure 5A). Half of these modules (*n* = 7) were strongly linked to the cell surface or mitochondrial compartments. The phenotype-related mitochondrial modules (M2, M4, M15, M20) all demonstrated decreased levels of expression in late disease, but notably displayed variable patterns of early disease expression (Figure 5B). For instance, while M4 and M15 appeared largely unchanged in AsymAD, M2 demonstrated a transient increase in AsymAD before plummeting in symptomatic disease. This transient pattern of M2 was particularly notable as it suggested possible pathways of AsymAD resilience related to intracellular energy processing. In contrast, M20 expression decreased significantly in preclinical disease and remained low in the symptomatic stage. M20, a small network with moderate ties to both the mitochondrial and ER compartments, included zeta-globulin (HBZ) among its hub proteins. Neuronal hemoglobin molecules, such as HBZ, play a critical role in maintaining mitochondrial function in the brain and have demonstrated altered levels in other neurodegenerative diseases [46,47].

The phenotype-related cell surface modules included M5, M8, and M9 (Figure 5C). M5, which demonstrated incremental increases in AsymAD and AD, was strongly linked to both the synapse and plasma membrane and boasted several ion transporters among its hub proteins. This included the α3 subunit of Na^+^/K^+^ ATPase (ATP1A3), in which mutations have been associated with several neurologic conditions, such as rapid-onset dystonia parkinsonism [48]. On the other hand, M8 and M9 both demonstrated decreased expression in early disease. Yet, while the levels of M8 further dropped in late AD, those of M9 remained largely stable from preclinical to symptomatic stages. These two modules featured many proteins involved in synaptic transmission, including key components of synaptic vesicles. The critical docking and fusion proteins VAMP2 and VAMP3 were both hubs of M8, while M9 harbored several other vesicular proteins, including SNAP25, synaptotagmin (SYT1), synaptogyrins (SYNGR1, SYNGR3), and SLC17A7. M8 also contained multiple proteins associated with the Golgi apparatus (ARFGAP1, RAB1A, RAB1B). Other membrane-associated modules related to clinical and pathological phenotypes in the AD brain included M16 (Figure 5D), M17 (Figure 5E), and M10 (not pictured), which localized strongly to the Golgi apparatus, ER, and nucleus (i.e., chromosome) respectively.

There were also three phenotype-related modules that localized strongly to the cytoskeleton/cytoplasm (Figure 5F). All three were significantly increased in late AD, but as with the mitochondrial modules, they demonstrated variable expression patterns in early disease. Two of these modules (M1, M23) demonstrated a clear, transient decrease in AsymAD before increasing dramatically in late disease. The third phenotype-associated cytoplasmic module was M3, which included APP and strongly mirrored CERAD scores in its gradual increases throughout disease. This module had secondary functional associations with protein import/targeting, suggesting that this group of proteins plays a prominent role in amyloid regulation. M7 was one of the few modules we resolved without strong ontological links to a cellular compartment. Yet, this module did display functional associations with G-protein receptor-mediated signaling and axonal guidance, suggesting links to both the cytoplasm and plasma membrane. While it remained stable in early disease, M7 increased significantly in late AD, consistent with its negative correlation to cognitive decline. Overall, these results demonstrated that many of the co-expression modules we resolved were linked to AD phenotypes and could play critical roles in both preclinical and symptomatic disease.

## 4. Discussion

In this proof-of-concept study, we applied a network-based proteomics approach to the analysis of membrane-fractionated brain samples from healthy control, AsymAD, and symptomatic AD cases. Our study yielded a sub-proteome derived from a wide array of membrane-bound compartments. In contrast to the robust cell type specificity that characterized the networks of our unfractionated brain analyses, the co-expression modules of this membrane proteome revealed little to no enrichment of cell type-specific markers and instead were organized principally by cellular compartmentalization. Nonetheless, we were able to link many of these modules strongly to AD diagnosis and associated clinicopathological phenotypes. Furthermore, many of these modules demonstrated notable changes in expression during preclinical disease stages. These results indicate that applying a systems-based analysis to the membrane sub-proteome could yield unique insights into the protein dynamics of early AD and its progression.

Above all, this approach and its compartment-driven modules could offer a valuable window into the complex intraorganellar processes of preclinical disease. In this analysis, we were able to resolve a level of detail in the protein expression patterns of certain membrane-bound compartments that was under-represented in the unfractionated proteome. For instance, our prior analyses of bulk brain homogenates have typically yielded one large mitochondrial co-expression module [9,10,11,12,13]. However, this study generated multiple modules with strong connections to the mitochondrial compartment. While nearly all of these mitochondrial modules decreased in late disease, consistent with hypometabolic phenotypes [49], we were able to observe variability in their expression patterns during the preclinical phase. This not only supports the notion that early mitochondrial changes are not uniformly hypometabolic, but also highlights the valuable role systems-based sub-proteomic analysis could play in unraveling the intricate organelle-specific protein alterations governing asymptomatic disease.

In similar fashion, this analysis also revealed notable heterogeneity among the preclinical expression patterns of our cell surface networks. All three of the phenotype-associated cell surface modules (M5, M8, and M9) either highly correlated to synaptic ontologies or contained multiple synaptic proteins. Yet, while M5 increased in early disease, M8 and M9 both decreased preclinically. Meanwhile, further investigation revealed that M9 fell to a stable level in AsymAD where it remained relatively unchanged in later disease, while M8 decreased in a more progressive fashion throughout both early and late AD. These results suggest that somewhat contrary to the uniform synaptic loss observed in the unfractionated network [9], there are possibly three groups of synaptic proteins altered in different ways during preclinical disease. That said, it is possible that other non-synaptic cell surface proteins were predominantly responsible for driving the variability in expression patterns among these modules. For instance, M5 also mapped heavily to non-specific plasma membrane ontologies and unlike M8 and M9, did not significantly overlap with the synaptic transmission module of the unfractionated proteome. This makes it difficult to draw general conclusions about synaptic dysfunction from M5 alone.

Notably, none of these AD-related cell surface modules (M5, M8, and M9) were significantly enriched in neuronal-specific markers. In fact, the current study indicates that fractionation may substantially obviate any sort of cell-specific network organization, an outcome that is perhaps intuitive given that this process is designed to separate the whole cell into its smaller, individual compartments. This ability to resolve cell type-independent networks presents a potentially useful strategy for further examining the nuances of the preclinical AD proteome, an entity free of the large changes in cell type abundance that tends to drive the proteomic results of symptomatic degeneration. Yet, while this proof-of-concept study successfully demonstrated the biological and clinical relevance of our experimental design, it is limited by its small sample size. Further examination of a larger set of membrane-fractionated samples is necessary to draw additional conclusions regarding the protein systems governing early AD and its progression.

## Figures and Tables

**Figure 1 proteomes-07-00030-f001:**
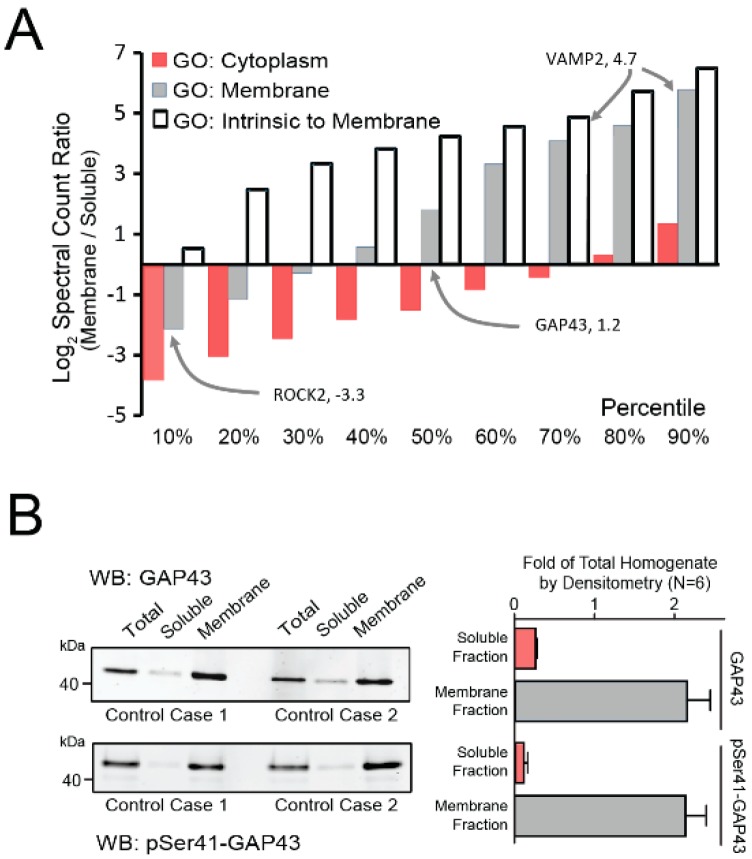
Characterization of Membrane-Enriched Samples. (**A**) Proteins identified by gene ontologies (GOs) as uniquely cytoplasmic, membrane, or intrinsic to membranes were quantified in both membrane and soluble fractions by peptide log_2_ spectral count ratio (membrane/soluble). Proteins were binned into deciles ranked by the log_2_(ratio) to represent the decile-specific average degree of enrichment or depletion within the membrane fraction. Only proteins with three or more peptide spectral counts were considered. VAMP2, a representative intrinsic membrane protein, was enriched in the membrane fraction, whereas the peripheral membrane protein ROCK2 was among proteins depleted in the membrane compared to the soluble fraction. The presynaptic protein GAP43 was also enriched in membrane fraction. (**B**) Western blots of total and phosphorylated (pSer41) GAP43 in total brain homogenate, soluble, and membrane fractions were performed on the 6 control samples. The left panel depicts the blot from two representative cases, while the right panel shows the quantified densitometry totals for all 6 cases. Enrichment of both phosphorylated and unmodified GAP43 was observed in the membrane fraction. Abbreviations: WB, Western Blot.

**Figure 2 proteomes-07-00030-f002:**
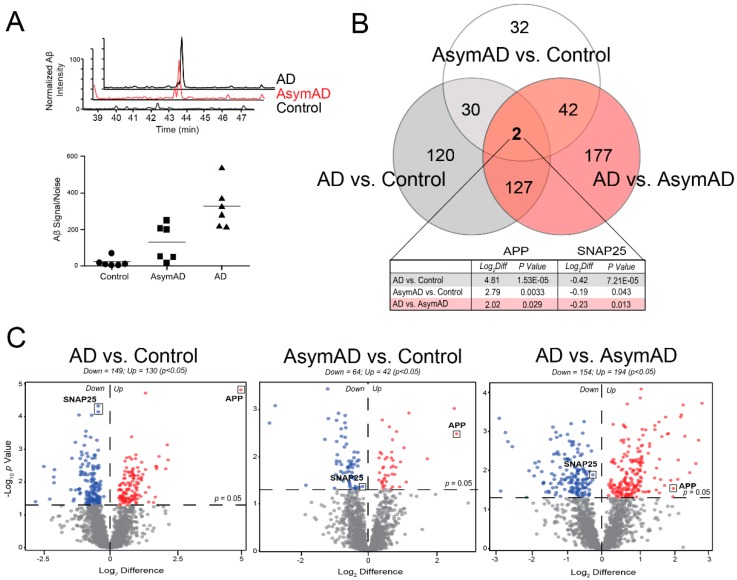
Differential Protein Abundance Across Disease Stages. (**A**) The top graph depicts extracted ion chromatograms for a fully tryptic amyloid precursor protein (APP) peptide corresponding to residues 17–28 of the Aβ sequence (LVFFAEDVGSNK) in a representative control, Asymptomatic Alzheimer’s Disease (AsymAD) and Alzheimer’s disease (AD) case. Signals were normalized by setting the maximum signal intensity of the AD sample to 100%. The bottom graph demonstrates the normalized peptide intensity of this Aβ sequence in all 18 cases. As expected, this measurement increased incrementally from control to AsymAD to AD cases. (**B**) Venn diagram for the 530 proteins significantly altered (*p* < 0.05) among the three pairwise comparisons, i.e., AD vs. Control, AsymAD vs. Control, and AD vs. AsymAD. APP and the synaptic protein SNAP25 were the only two proteins to demonstrate significant changes in all pairwise comparisons. (**C**) Volcano plots display the log-transformed fold change (Log_2_ Difference) against the log-transformed Tukey-adjusted ANOVA *p* value (−Log_10_
*p* Value) for all proteins of each pairwise comparison. Those proteins with significantly decreased expression (*p* < 0.05) for each comparison are shown in blue, while the proteins with significantly increased expression (*p* < 0.05) are noted in red. Abbreviations: AD, Alzheimer’s disease; AsymAD, Asymptomatic Alzheimer’s Disease; Log_2_Diff, Log_2_ Difference (i.e., Log_2_ Fold Change).

**Figure 3 proteomes-07-00030-f003:**
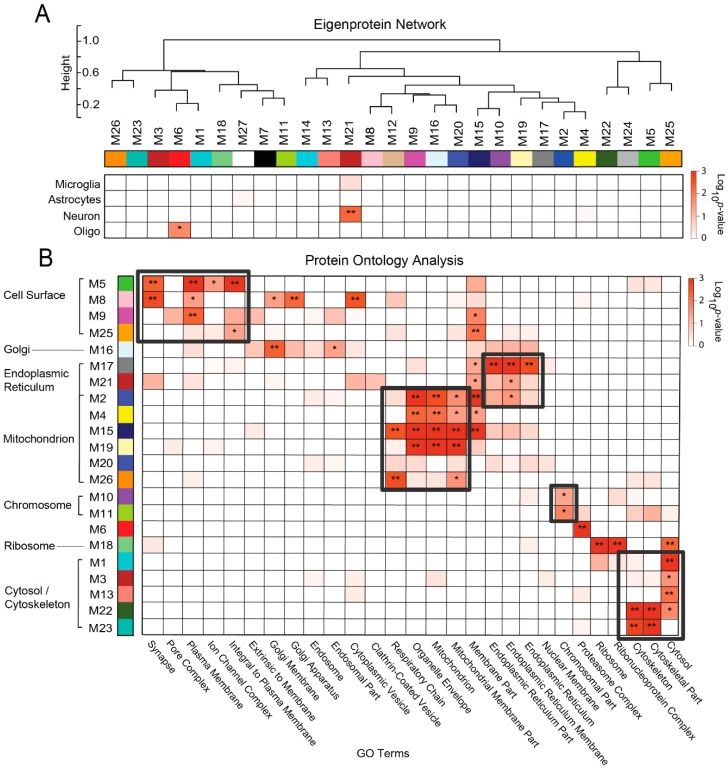
Network Modules Correlate to Membrane-Bound Cellular Compartments. (**A**) Weighted protein co-expression network analysis (WPCNA) grouped proteins (*n* = 1808) into distinct protein modules (M1–M27) that were then clustered to assess module relatedness based on correlation of protein co-expression eigenproteins. A hypergeometric Fisher exact test revealed only two networks with significant enrichment of cell-type specific markers (* *p* < 0.05; ** *p* < 0.01). (**B**) A separate Fisher exact test demonstrated strong module associations with human gene ontologies related to membrane-bound cellular compartments (* *p* < 0.05; ** *p* < 0.01). There were six modules (M2, M4, M15, M19, M20, M26) that correlated most strongly and/or specifically with gene ontologies related to the mitochondrion or mitochondrial membrane. In contrast, there were four modules (M5, M8, M9, M25) with strong correlations to synaptic/cell surface terms. Other membrane-bound compartments highly represented in this proteome included the endoplasmic reticulum (M2, M17, M21), nucleus (i.e., chromosome) (M10, M11), and Golgi apparatus (M8, M16). Finally, five modules were highly linked to the cytosol/cytoskeleton (M1, M3, M13, M22, M23). Abbreviations: M, Module; GO, Gene Ontology.

**Figure 4 proteomes-07-00030-f004:**
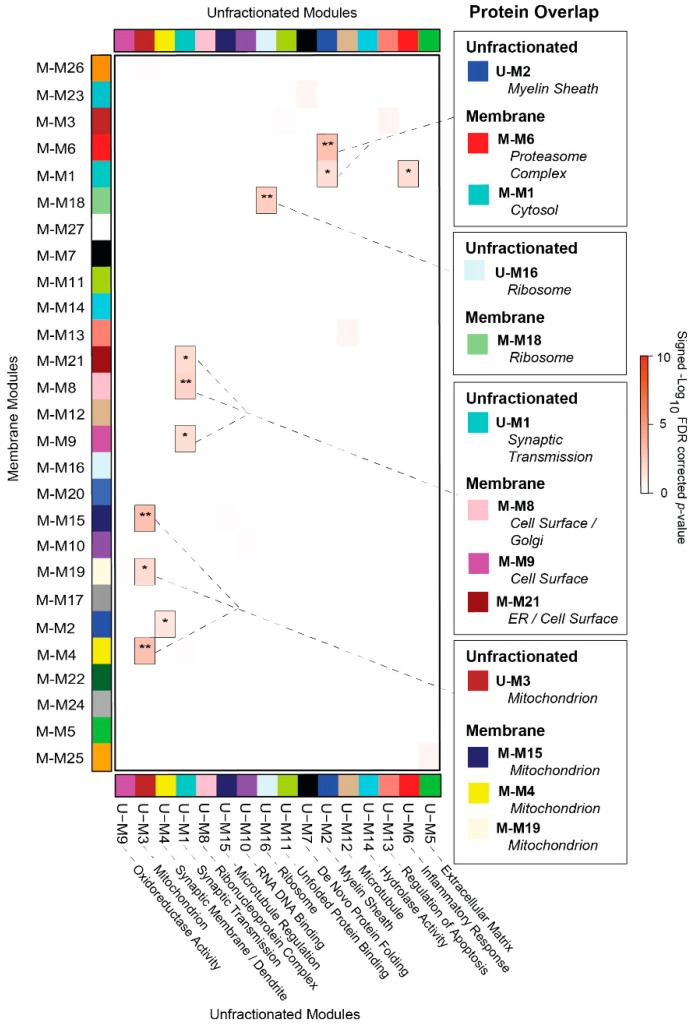
Unfractionated and Membrane-Associated Co-Expression Networks Demonstrate Minimal Overlap. A hypergeometric one-tailed Fisher’s exact test (FET) was used to identify modules that shared significant overlap of protein members between the membrane-fractionated (M) network and that of unfractionated control, AsymAD, and AD cases derived from the Baltimore Longitudinal Study of Aging (BLSA). The 16 modules of the unfractionated (U) BLSA network, clustered by eigenprotein relatedness, are shown on the x-axis along with their top protein ontologies. These BLSA modules were aligned to the 27 modules of the membrane-associated network (y-axis). Module gene symbol lists showed either significant overlap (red) or no significant under- or over-representation (white) in protein membership. Numbers are positive signed −Log_10_(FDR-corrected p values) representing the degree of overlap (* *p* < 0.05; ** *p* < 0.01). Notable overlapping modules are highlighted to the right of the FET results.

**Figure 5 proteomes-07-00030-f005:**
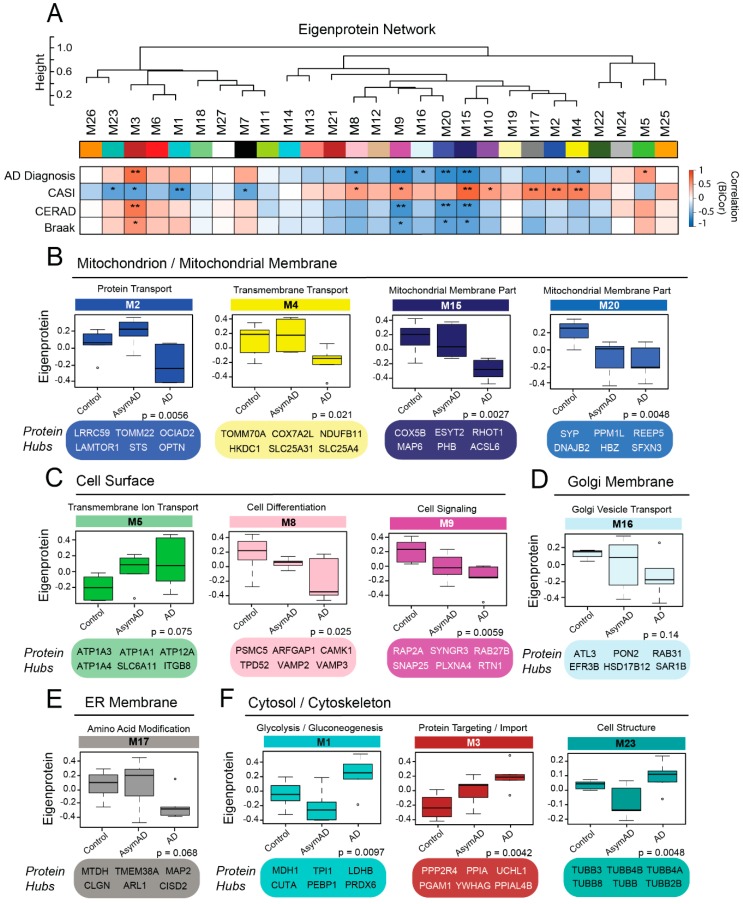
Modules in the Membrane Proteome are Correlated to Clinical and Pathological AD Phenotypes. (**A**) Biweight midcorrelation (bicor) analysis of module co-expression eigenproteins to clinical and pathological disease traits, including AD diagnosis, cognitive decline as measured by Cognitive Assessment Screening Instrument (CASI) score, and cortical levels of amyloid (Consortium to Establish a Registry for Alzheimer’s Disease (CERAD) score) and tau (Braak score). There were 14 modules with significant correlations to one or more disease-associated traits (* *p* < 0.05; ** *p* < 0.01). (**B**–**F**) Module expression profiles and key hub proteins of trait-associated modules organized by compartment localization (GO terms). *p* values were calculated for each expression profile using Kruskal–Wallis one-way nonparametric ANOVA. Abbreviations: AD, Alzheimer’s disease; AsymAD, Asymptomatic Alzheimer’s Disease; ER, Endoplasmic Reticulum.

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
