# Peer review of "Network Analysis of a Membrane-Enriched Brain Proteome across Stages of Alzheimer’s Disease"

_proteomes, 2019, doi:10.3390/proteomes7030030_

Round 1

Reviewer 1 Report

Here the authors sought to investigate co-regulation networks built from MS analyses of human brain membrane enrichments from control, asymptomatic AD, and symptomatic AD. In this proof of concept study, they sought to answer two questions: 1. Do these membrane enrichment co-regulation networks differ from co-expression networks in an unfractionated proteome and 2. Do membrane enrichment protein levels differ between the three groups. This group is one of the leaders in the field and as expected the biochemistry, MS, bioinformatic, and statistical analyses have been conducted in a highly rigorous fashion.  That said, the authors may have been better served by addressing one or the other of these questions. The control – asymptomatic AD – symptomatic AD comparisons are underpowered (as the authors acknowledge) and the network analysis maybe confounded if co-regulation modules are not preserved across the different groups. The authors note that co-expression modules are preserved in their previous analysis of unfractionated tissue, but this may not be the case for membrane fractions from AD tissue. Nevertheless, the findings seem highly plausible, are broadly in line with prior studies, and this is a relatively minor issue for a proof of concept study.

The larger issue is the use of data from their previous study on an unfractionated proteome as a comparison; as opposed to investigating the unfractionated proteome from these 18 subjects. This prior study utilized a different cohort, digestion approach, and instrumentation… and ultimately would have generated networks from a different group of proteins. While it is unlikely that the top-line results would change (the observation of cell-type specific modules in unfractionated tissue vs membrane compartment specific modules in the membrane enrichments), this omission represents a missed opportunity to more fully compare the fractionated and unfractionated proteomes. Specifically, it would be interesting to determine if there are a. modules that are preserved between the two proteomes and/or b. modules that are correlated between the two proteomes.

Additional minor issues:

1.       Why were the membrane fractions taken up in urea if they were to be fractionated and digested on gel.

2.       Several other groups (S. Grant, C.G. Hahn, J. Meador-Woodruff) have validated the stability of various membrane fractions (synaptosomes, vesicles, PSDs) in human postmortem brain tissue for MS. It would seem critical to site this body of work in making the case for systems level analyses of enriched proteomes in human brain tissue as PMI is not directly investigated in this study.

Reviewer 2 Report

1. Figure S1B should indicate the significant difference on the bands, which will make it easier for readers to read.

2. "The alkylated samples were separated on a 10% SDS gel and stained with Coomassie Blue G-250. Each sample lane was cut into five gel bands corresponding to molecular weight ranges to increase the depth of coverage of the proteome." Please added this figure in the manuscript.

3. "Data dependent acquisition of centroid MS spectra at 30,000 resolution and MS/MS  spectra were obtained in the LTQ following collision induced dissociation (collision energy 35%, activation Q 0.25, activation time 30 ms) for the top 10 precursor ions with charge determined by the acquisition software to be z ≥ 2." Why the authors set the top 10 precursor ions? It was usually set to 6.

4. For the database search, why did the authors SEQUEST 3.5? I think it is more popular by using MASCOT.

5. It is not easy to understand how the authors select the unique proteins by proteomic approaches.

For the protein identification, how to make sure it is not a false identification? Did the authors perform any validation for the quality of MS/MS spectra?

6. More discussion for Immunoblotting should be added in the manuscript.

7. "Our study yielded a deep sub-proteome derived from a wide array of membrane-bound compartments. In contrast to the robust cell type specificity that characterized the networks of our unfractionated brain analyses, the co-expression modules of this membrane proteome revealed little to no enrichment of cell type-specific markers and instead were organized principally by cellular compartmentalization. Nonetheless, we were able to link many of these modules strongly to AD diagnosis and associated clinicopathological phenotypes. Furthermore, many of these modules demonstrated notable changes in expression during preclinical disease stages." How to connect to the clinical diagnosis? If the authors can provide the data of pathological diagnosis, it will be helpful.
